# A Lightweight 1-D Convolution Augmented Transformer with Metric Learning for Hyperspectral Image Classification

**DOI:** 10.3390/s21051751

**Published:** 2021-03-03

**Authors:** Xiang Hu, Wenjing Yang, Hao Wen, Yu Liu, Yuanxi Peng

**Affiliations:** The State Key Laboratory of High-Performance Computing, College of Computer, National University of Defense Technology, Changsha 410000, China; huxiang@nudt.edu.cn (X.H.); wenjing.yang@nudt.edu.cn (W.Y.); hao.wen@nudt.edu.cn (H.W.); liuyu11@nudt.edu.cn (Y.L.)

**Keywords:** deep learning, transformer, metric learning, 1-D convolution, hyperspectral image classification, remote sensing

## Abstract

Hyperspectral image (HSI) classification is the subject of intense research in remote sensing. The tremendous success of deep learning in computer vision has recently sparked the interest in applying deep learning in hyperspectral image classification. However, most deep learning methods for hyperspectral image classification are based on convolutional neural networks (CNN). Those methods require heavy GPU memory resources and run time. Recently, another deep learning model, the transformer, has been applied for image recognition, and the study result demonstrates the great potential of the transformer network for computer vision tasks. In this paper, we propose a model for hyperspectral image classification based on the transformer, which is widely used in natural language processing. Besides, we believe we are the first to combine the metric learning and the transformer model in hyperspectral image classification. Moreover, to improve the model classification performance when the available training samples are limited, we use the 1-D convolution and Mish activation function. The experimental results on three widely used hyperspectral image data sets demonstrate the proposed model’s advantages in accuracy, GPU memory cost, and running time.

## 1. Introduction

Hyperspectral image (HSI) classification is a focus point in remote sensing because of its many uses across fields, such as change area detection [1], land-use classification [2,3], and environmental protection [4]. However, because of redundant spectral band information, large data size, and a limited number of training samples, the pixel-wise classification of the hyperspectral image remains a formidable challenge.

Deep learning (DL) has become extremely popular because of its ability to extract features from raw data. It has been applied in computer vision tasks, such as image classification [5,6,7,8], object detection [9], semantic segmentation [10], and facial recognition [11]. As a classical visual classification task, HSI classification has also been influenced by DL. For example, Chen et al. [12] proposed a stacked autoencoder for feature extraction. Ma et al. [13] introduced an updated deep auto-encoder to extract spectral–spatial features. Zhang et al. [14], adopted a recursive autoencoder (RAE) as a high-level feature extractor to produce feature maps from the target pixel neighborhoods. Chen et al. [15], combined deep belief network (DBN) and restricted Boltzmann machine (RBM) for hyperspectral image classification.

However, these methods extract the features with destroying the initial spatial structure. Because convolutional neural networks (CNNs) can extract spatial features without destroying the original structure, new methods based on CNNs have been introduced. For example, Chen et al. [16] designed a novel 3-D-CNN model with regularization for HSI classification. Roy et al. [17] proposed a hybrid 3-D and 2-D model for HSI classification.

In addition to modifying the CNN structure, deep metric learning (DML) has been applied to improve CNN classification performance. The metric learning loss term was introduced to the CNN objective function to enhance the model’s discriminative power. Cheng et al. [18] designed a DML method based on existing CNN models for remote sensing image classification. Guo et al. [19] proposed a DML framework for HSI spectral–spatial feature extraction and classification.

Recently, another deep learning model, the transformer, has been applied for computer vision tasks. In Reference [20], transformers were proposed for machine translation and became the state-of-art model in many natural language processing (NLP) tasks. In Reference [21], the transformer network’s direct application, Vision Transformer, to image recognition was explored.

In this paper, inspired by the Vision Transformer, we propose a lightweight network based on the transformer for hyperspectral image classification. The main contributions of the paper are described below.

(1) First, the key part of our proposed model is the transformer encoder. The transformer encoder does not use convolution operations, requiring much less GPU memory and fewer trainable parameters than the convolutional neural network. The 1-D convolution layer in our model serves as the projection layer to get the embedding of each sequence.

(2) Second, to get a better classification performance, we replace the linear projection layers in the traditional vision transformer model for computer vision with the 1-D convolution layer and adopt a new activation function, Mish [22].

(3) Third, we introduce the metric learning mechanism, which makes the transformer model more discriminative. We believe the present study is the first to combine the metric learning and the transformer model in hyperspectral image classification.

The rest of this article is organized as follows. Section 2 introduces the proposed framework. The experimental results and analysis of different methods are provided in Section 3 and Section 4. Finally, Section 5 presents the conclusions.

## 2. Methods

The overall architecture of our proposed model is demonstrated in Figure 1. Firstly, we split the input image into fixed-size patches and reshaped them into 1-D sequences. Next, we use a 1-D convolution layer to get the embedding of each sequence. The embedding of the central sequence will be supervised by center loss. After adding the position embedding to the sequence, the sequences will be fed to a standard two-layer Transformer encoder. The fully connected layers will handle the result. The fully connected layers consist of a layernorm layer, several fully connected layers, and Mish activation function. The output is the classification result.

### 2.1. Projection Layer with Metric Learning

We use the 1-D convolution layer as the projection layer. The 1-D convolution is calculated by convolving a 1-D kernel with the 1-D-data. The computation complexity of the 1-D convolution layer is drastically lower than the 2-D and 3-D convolution layers. It leads to a significant advantage of running time for 1-D convolution layers over 2-D and 3-D convolution layers. The computational process is presented in Figure 2. In 1-D convolution, the activation value at spatial position *x* in the *j*th feature map of the *i*th layer, denoted as $v_{i,j}^{x}$, is generated using Equation (Equation 1) [23].
(1)$$v_{i,j}^{x}=b_{i,j}^{x}+\sum_{m}\sum_{r=0}^{R_{i}-1}W_{i,j,m}^{r}V_{i-1,m}^{x+r},$$
where *b* is the bias, *m* denotes the feature cube connected to the current feature cube in the (i−1)th layer, *W* is the *r*th value of the kernel connected to the *m*th feature cube in the prior layer, and *R* denotes the length of the convolution kernel size.

In our model, the input image is split into 25 patches. So, we apply 25 projection layers. In order to make the model perform better, we adopt two techniques, parameter sharing, and metric learning. In our proposed model, the parameter sharing strategy can accelerate the model convergence rate and promote the classification accuracy.

The metric learning can enhance the discriminative power of the model by decreasing the intraclass distances and increasing the interclass distances. The metric learning loss term that we use in our experiment is the center loss [24], formulated as:(2)$$L_{C}=\frac{1}{2M}\sum_{i=1}^{M}\left|\right|x_{i}^{*}-c_{y_{i}}{\left|\right|}_{2}^{2}$$
where *x${}_{i}^{*}$* denotes the learned embedding of the *i*th input central patch in the batch, for *i* = 1, ⋯, *M*, and *c${}_{k}$* is the *k*th class center based on the embeddings in *k*th class, for *k* = 1, ⋯, *K*.

### 2.2. Transformer Encoder

The transformer network was proposed in Vaswani et al. [20]. It is composed of several identical layers. Each layer was made up of two sub-layers, the multi-head self-attention mechanism and the fully connected feed-forward network, as shown in Figure 3. A residual connection followed by layer normalization is employed in each sub-layer. So, the output of each sub-layer can be defined as *LayerNorm(x + SubLayer(x))*, where *SubLayer(x)* denotes the function implemented by the sub-layer. The multi-head self-attention is defined as:(3)$$\;\mathrm{MultiHead}\;\left(Q,K,V\right)=\mathrm{concat}\left(\mathrm{head}{\;}_{1},\cdots ,\mathrm{head}{\;}_{h}\right)W^{O},$$
where head${}_{i}=$ Attention $\left(QW_{i}^{Q},KW_{i}^{K},VW_{i}^{V}\right),W_{i}^{Q}\in R^{d_{\mathrm{model}\;}\times d_{q}}$, $W_{i}^{K}\in R^{d_{\mathrm{model}\;}\times d_{k}}$, $W_{i}^{V}\in R^{d_{\mathrm{model}\;}\times d_{v}}$, and $W^{O}\in R^{h\times d_{v}\times d_{\mathrm{model}\;}}$ are parameter matrices. The attention is formulated as:(4)$$\;\mathrm{Attention}\;\left(Q,K,V\right)=\mathrm{softmax}\left(\frac{QK^{T}}{\sqrt{d_{k}}}\right)V,$$
where *Q*, *K* of dimension $d_{k}$, and *V* of dimension $d_{v}$ are three defined learnable weight matrices.

### 2.3. Fully Connected Layers

The fully connected layers consist of a layernorm layer, two fully connected layers, and the Mish activation function. The Mish function is defined as
(5)$$Mish\left(x\right)=x\times tanh(softplus(x))=x\times tanh(ln\left(1+e^{x}\right)),$$
where *x* is the input of the function. The difference between Mish and Relu is shown in Figure 4. The benefit of Mish will be proved in the next section.

## 3. Experiment

### 3.1. Data Set Description and Training Details

We evaluate the proposed model on three publicly available hyperspectral image data sets, namely Indian Pines, University of Pavia, and Salinas, as illustrated in Figure 5, Figure 6 and Figure 7. The spectral radiance of these three data sets and the corresponding categories are shown in Figure 8, Figure 9, Figure 10 and Figure 11.

The Indian Pines data set was collected by the Airborne Visible/Infrared Imaging Spectrometer (AVIRIS) from Northwestern Indiana. The scene contains 145 ×145 pixels with a spatial resolution of 20 m by pixel and 224 spectral channels in the wavelength range from 0.4 to 2.5 μm. After 24 bands corrupted by water absorption effects were removed, 200 bands were available for analysis and experiments. The 10,249 labeled pixels were divided into 16 classes.

The University of Pavia data set was acquired by the Reflective Optics System Imaging Spectrometer (ROSIS) over Pavia, northern Italy. The image size was 610 ×340 with a spatial resolution of 1.3 m by pixel and 103 spectral bands in the wavelength range from 0.43 to 0.86 μm. The 42,776 labeled pixels were designed into 9 categories.

The Salinas data set was gathered by the AVIRIS sensor over Salinas Valley, California. The image contains 512 ×217 pixels with a spatial resolution of 3.7 m by pixel and 224 spectral channels in the wavelength range from 0.4 to 2.5 μm. After 20 water-absorbing spectral bands were discarded, 204 bands were accessible for classification. The 54,129 labeled pixels were partitioned into 16 categories.

For the Indian Pines, the proportion of samples for training and validation was set to 3%. For the University of Pavia, we set the ratios of training samples and validation samples to 0.5%. For the Salinas, we selected 0.4% of the samples for training and 0.4% for validation. Table 1, Table 2 and Table 3 list the number of training, validation, and testing samples of the three data sets.

All experiments were conducted using the same device with the RTX Titan GPU and 16 GB RAM. The learning rate was set to 0.0005, and the number of epochs was 200. The model with the least cross-entropy loss in the validation samples was selected for testing. We used mini-batches with a size of 256 for all experiments. The weight of the metric learning loss term was set to 10${}^{-6}$. We apply the traditional principal component analysis (PCA) over the original HSI data to remove the spectral redundancy. To compare the performances of experimental models fairly, we use the same input size over each data set, such as 25 × 25 × 30 for the Indian Pines and 25 × 25 × 15 for the University of Pavia and the Salinas, respectively. In order to ensure the accuracy and stability of the experimental results, we conducted the experiments 10 times consecutively. The parameter summary of the proposed transformer model architecture over the three data sets is shown in Table 4, Table 5, Table 6 and Table 7.

### 3.2. Classification Results

To evaluate the performance of the proposed method, we used three metrics: overall accuracy (OA), average accuracy (AA), and the Kappa coefficient (Kappa). OA is the percentage of correctly classified samples over the total test samples. AA is the average accuracy from each class, and Kappa measures the consistency between the predicted result and ground truth. The results of the proposed transformer model are compared with traditional deep-learning methods, such as 2-D-CNN [25], 3-D-CNN [26], Multi-3-D CNN [27], and hybridSN [17].

Table 8, Table 9 and Table 10 show the categorized results using different methods. The numbers after the plus-minus signs are the variances of the corresponding metrics. Figure 12, Figure 13 and Figure 14 demonstrate classification maps for each of the methods. The preliminary analysis of the results revealed that our proposed model could provide a more accurate classification result than other models over each data set. Among the contrast models, the OA, AA, and Kappa of HybridSN were higher than those of other contrast models. It indicates that the 3-D-2-D-CNN models were more suitable for the hyperspectral image classification with limited training samples than the models that used 2-D convolution or 3-D convolution alone. Secondly, it can be observed from these results that the performance of 2-D-CNN was better than that of 3-D-CNN or Multi-3-D-CNN. To the best of our knowledge, it was probably because the large parameter size can easily lead to overfitting when the training samples were lacking.

Considering the spectral information, we can conclude that the spectral information can influence the classification accuracy greatly. For example, in Indian Pines, the spectral feature of Grass-pasture-mowed is significantly different from the features of the other classes over the first 75 spectral bands. Besides, the pixels of Grass-pasture-mowed have similar spectral features. Although we only used one sample of Grass-pasture-mowed for training. All the models can reach the accuracy of at least 85%.

Table 11 summarizes the parameter size of the five models. The two rows of each model are the number of parameters and the memory space occupied by the parameters. It is apparent from this table that the transformer model was of the smallest parameter size, indicating the efficiency of the transformer model.

Table 12 shows the floating-point operations (flops) of the five models. Table 13, Table 14 and Table 15 compare the training time and testing time of the five models over each data set. The computational costs of our proposed model were the least. Because of the GPU acceleration for convolutions, the 2-D-CNN was quicker than our proposed model. Considering that the accuracy of the 2-D-CNN was lower than that of our proposed model, we think our method has a better balance of accuracy and efficiency.

## 4. Discussion

In this part, further analysis of our proposed model is provided. Firstly, metric learning can improve the model classification performance significantly, especially when the training samples are extremely lacking, and the results prove it. Secondly, the results of controlled experiments reflect the benefits of 1-D convolution layers with parameter sharing. Thirdly, the experimental results about different activation functions confirm the superiority of Mish.

### 4.1. Effectiveness of the Metric Learning

To prove the effectiveness of the metric learning, we remove the metric learning mechanism and compare the performance between these two models.

Table 16 and Figure 15, Figure 16 and Figure 17 reveal the benefits of the metric learning mechanism. The numbers after the plus-minus signs are the variances of the corresponding metrics. The model with metric learning can reach a higher accuracy. We can conclude that the metric learning can improve the model classification results.

### 4.2. Effectiveness of the 1-D Convolution and Parameter Sharing

In Section 2.1, we declare that the parameter sharing strategy can improve the model classification result. Here, we will compare the performance of the transformer model with the 1-D convolution and parameter sharing, the transformer model with 1-D convolution, the transformer model with the linear projection layers and parameter sharing, and the transformer model with the linear projection layers.

From Figure 18, we can conclude that both 1-D convolution layers and parameter sharing boost the model performance.

### 4.3. Effectiveness of the Activation Function

The Mish activation function can promote model performance slightly. Figure 19 shows the classification OA of the transformer models based on different activation functions.

## 5. Conclusions

In this article, we introduce the transformer architecture for hyperspectral image classification. Meanwhile, by replacing the linear projection layer with the 1-D convolution layer, the image patches can be embedded into sequences with more information. It can lead to an increase in classification accuracy. Besides, the Mish activation function is adopted instead of the Relu activation function; hence, the model performance can be further boosted.

In the experiments, the influence of three innovative changes based on the classical vision transformer, including metric learning, the 1-D convolution layer, and the Mish activation function, is proved. Moreover, many state-of-the-art methods based on convolutional neural networks, including 2-D CNN, 3-D CNN, multi-scale 3-D CNN, and hybrid CNN, are taken into account. The results demonstrate the advantage of the proposed model, especially under the condition of lacking training samples.

## Figures and Tables

**Figure 1 sensors-21-01751-f001:**
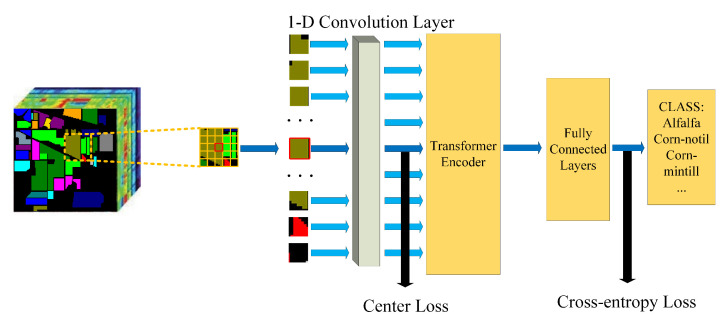
The overall architecture of our proposed model.

**Figure 2 sensors-21-01751-f002:**
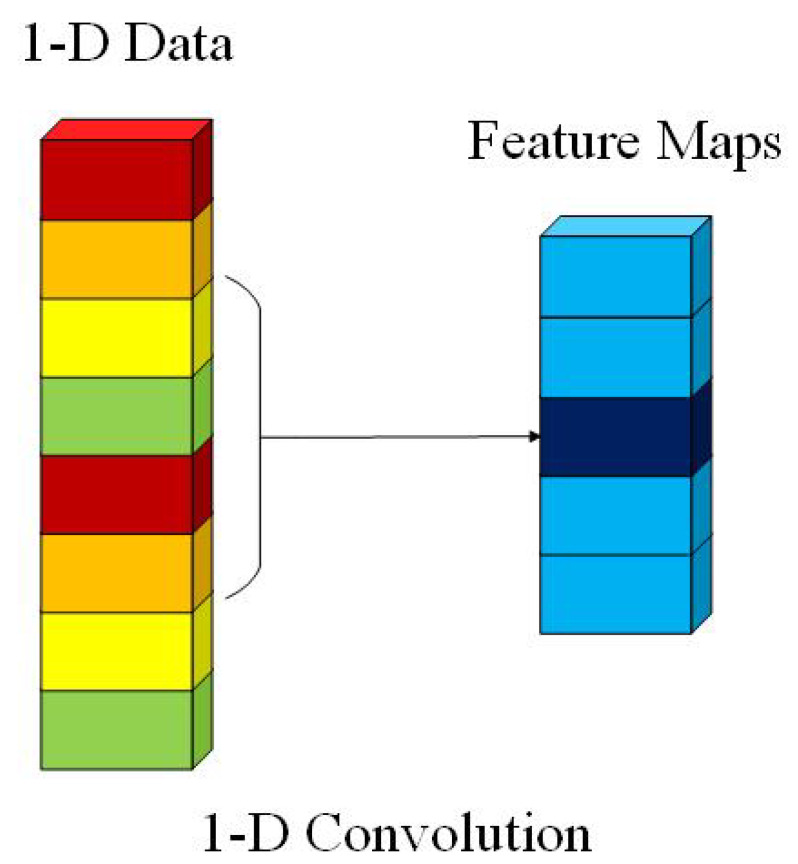
The computational process of 1-D convolution.

**Figure 3 sensors-21-01751-f003:**
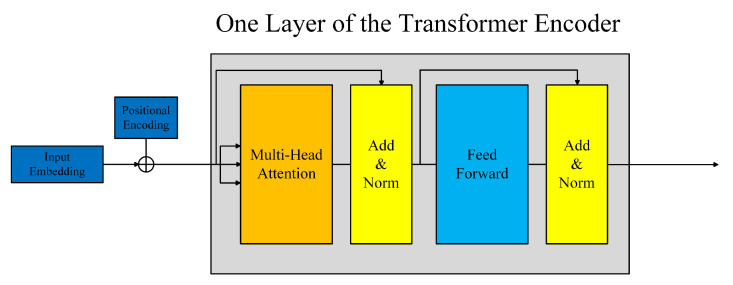
The architecture of the transformer encoder.

**Figure 4 sensors-21-01751-f004:**
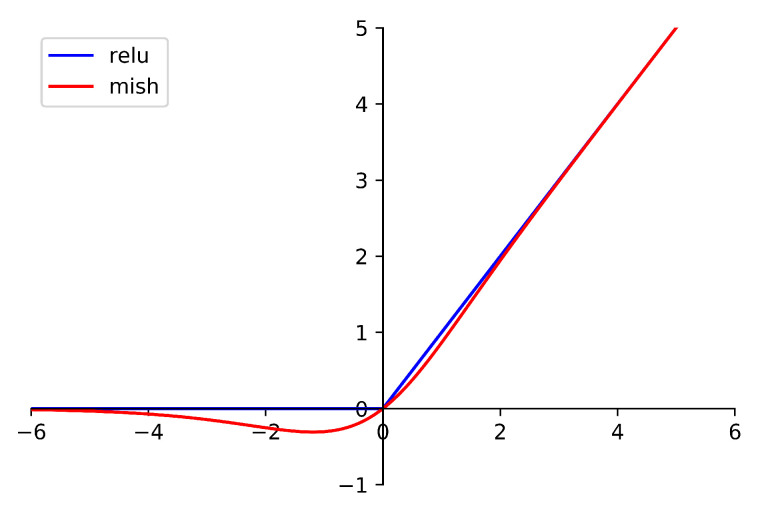
The difference between Mish and Relu.

**Figure 5 sensors-21-01751-f005:**
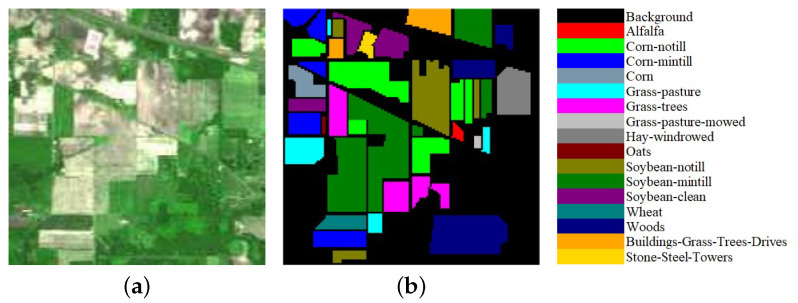
(**a**) False-color Indian Pines image. (**b**) Ground-truth map of the Indian Pines data set.

**Figure 6 sensors-21-01751-f006:**
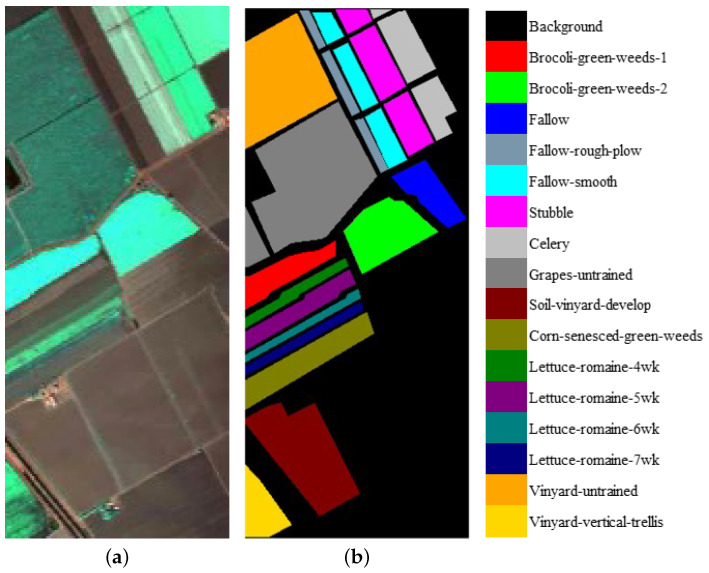
(**a**) False-color Salinas image. (**b**) Ground-truth map of the Salinas data set.

**Figure 7 sensors-21-01751-f007:**
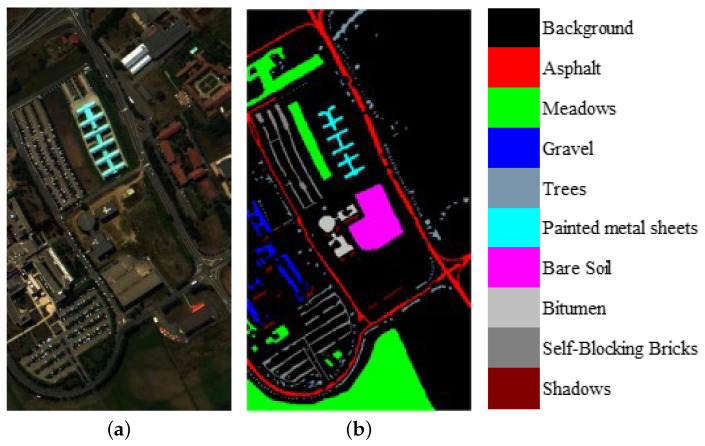
(**a**) False-color University of Pavia image. (**b**) Ground-truth map of the University of Pavia data set.

**Figure 8 sensors-21-01751-f008:**
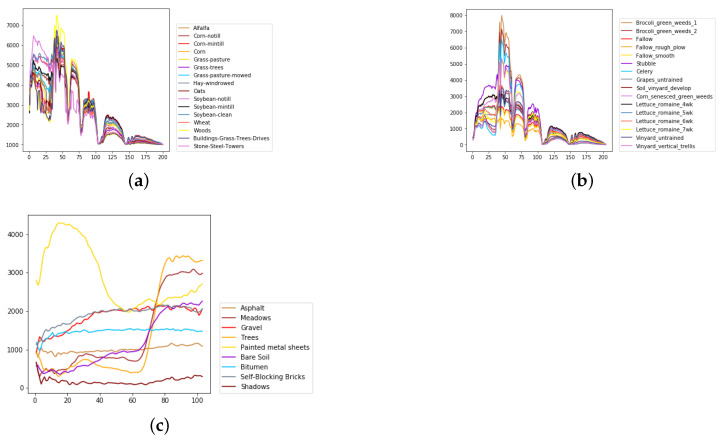
The overall spectral radiance and the corresponding categories in different data sets. (**a**) Indian Pines. (**b**) Salinas. (**c**) University of Pavia.

**Figure 9 sensors-21-01751-f009:**
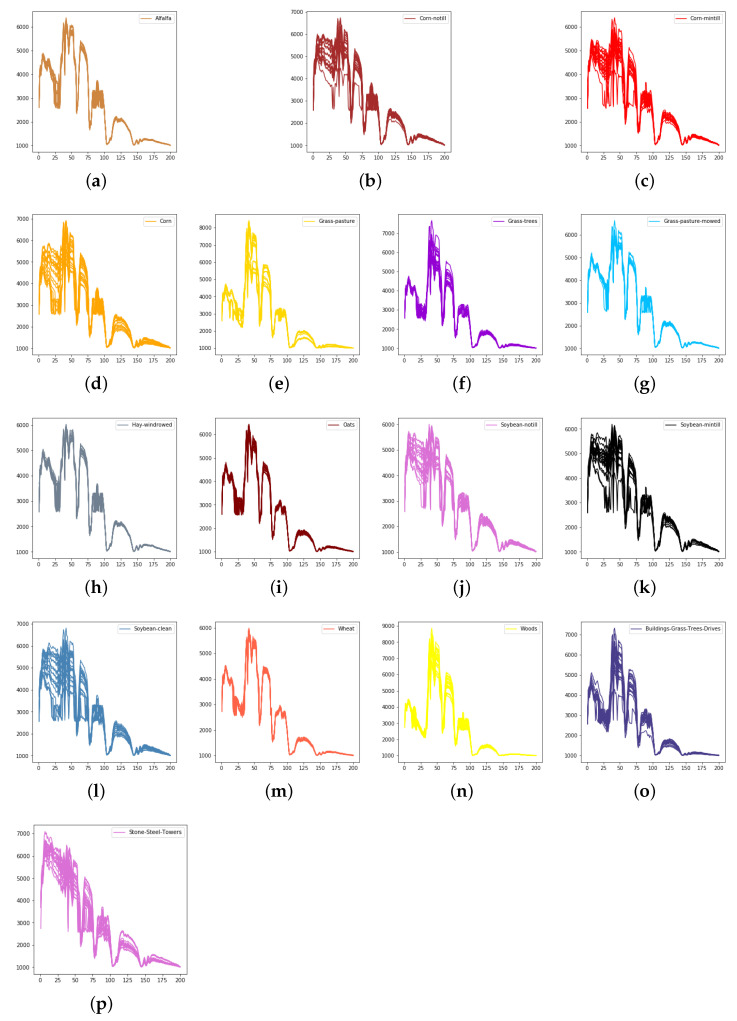
The spectral radiance of different pixels and the corresponding categories in Indian Pines. (**a**) Alfalfa. (**b**) Corn-notill. (**c**) Corn-mintill. (**d**) Corn. (**e**) Grass-pasture. (**f**) Grass-trees. (**g**) Grass-pasture-mowed. (**h**) Hay-windrowed. (**i**) Oats. (**j**) Soybean-notill. (**k**) Soybean-mintill. (**l**) Soybean-clean. (**m**) Wheat. (**n**) Woods. (**o**) Buildings-Grass-Trees-Drives. (**p**) Stone-Steel-Towers.

**Figure 10 sensors-21-01751-f010:**
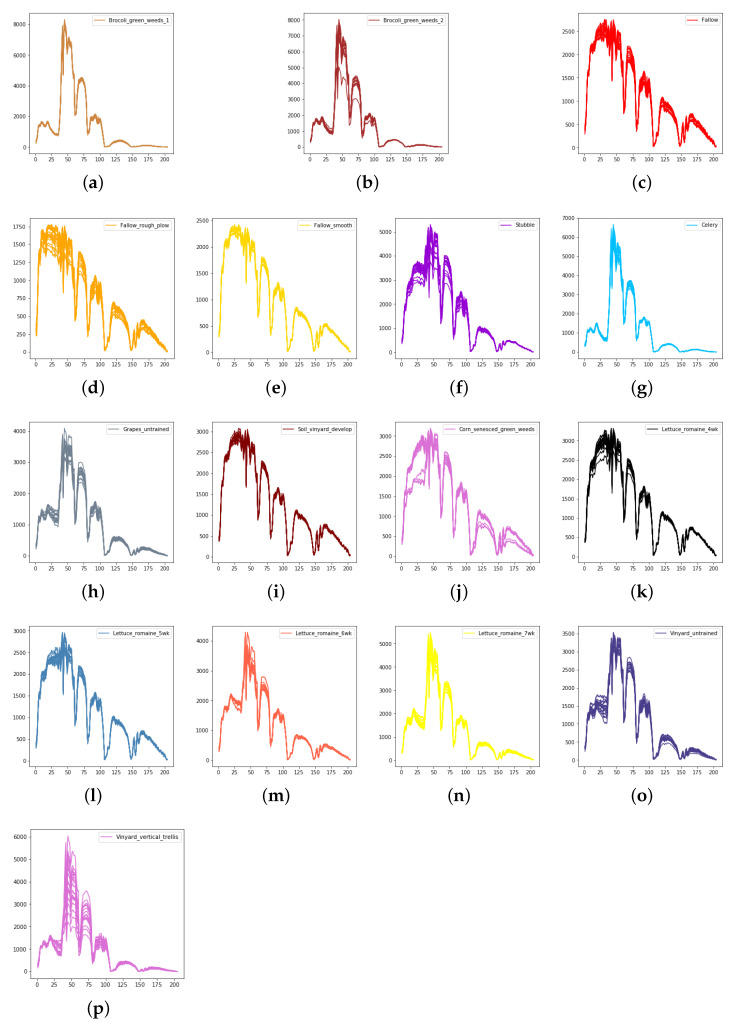
The spectral radiance of different pixels and the corresponding categories in Salinas. (**a**) Brocoli_green_weeds_1. (**b**) Brocoli_green_weeds_2. (**c**) Fallow. (**d**) Fallow_rough_plow. (**e**) Fallow_smooth. (**f**) Stubble. (**g**) Celery. (**h**) Grapes_untrained. (**i**) Soil_vinyard_develop. (**j**) Corn_senesced_green_weeds. (**k**) Lettuce_romaine_4wk. (**l**) Lettuce_romaine_5wk. (**m**) Lettuce_romaine_6wk. (**n**) Lettuce_romaine_7wk. (**o**) Vinyard_untrained. (**p**) Vinyard_vertical_trellis.

**Figure 11 sensors-21-01751-f011:**
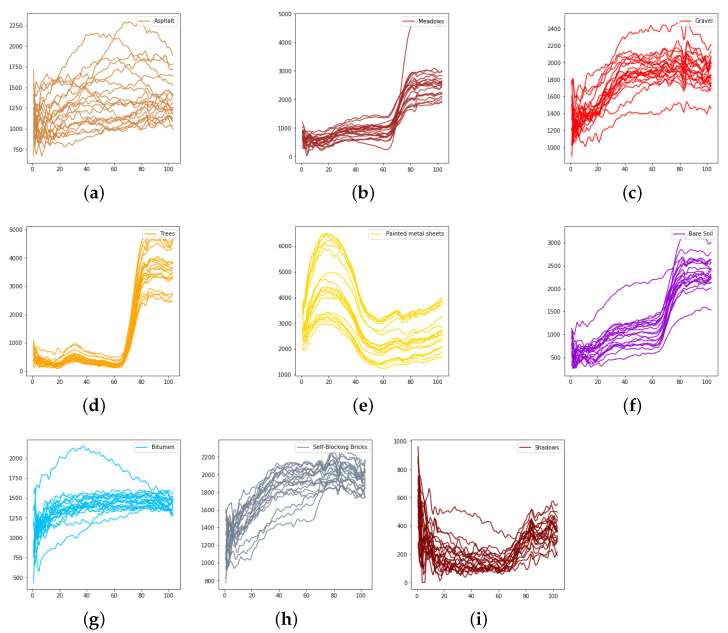
The spectral radiance of different pixels and the corresponding categories in the University of Pavia. (**a**) Asphalt. (**b**) Meadows. (**c**) Gravel. (**d**) Trees. (**e**) Painted metal sheets. (**f**) Bare Soil. (**g**) Bitumen. (**h**) Self-Blocking Bricks. (**i**) Shadows.

**Figure 12 sensors-21-01751-f012:**
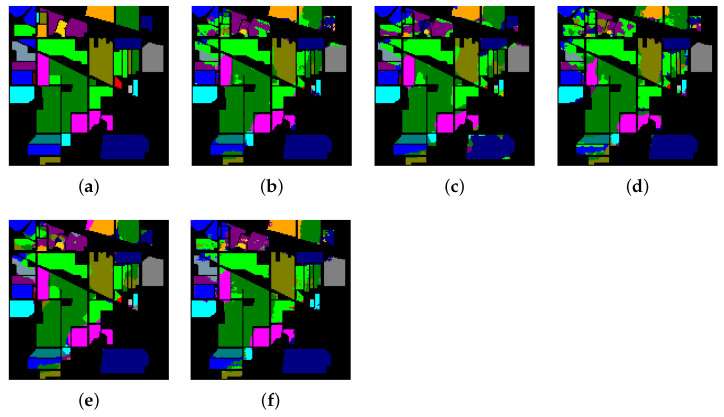
The classification maps of Indian Pines. (**a**) Ground-truth map. (**b**)–(**f**) Classification results of 2-D-convolutional neural network (CNN), 3-D-CNN, Multi-3-D-CNN, HybridSN, and Transformer.

**Figure 13 sensors-21-01751-f013:**
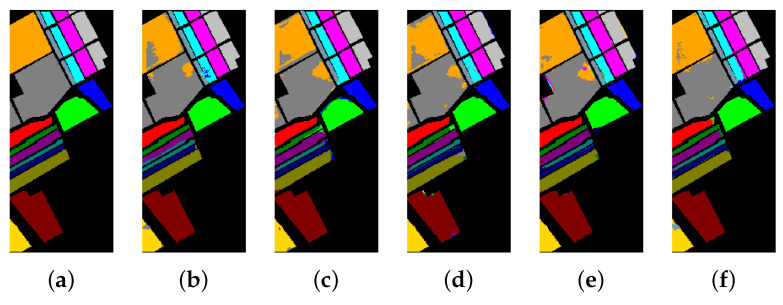
The classification maps of Salinas. (**a**) Ground-truth map. (**b**)–(**f**) Classification results of 2-D-CNN, 3-D-CNN, Multi-3-D-CNN, HybridSN, and Transformer.

**Figure 14 sensors-21-01751-f014:**
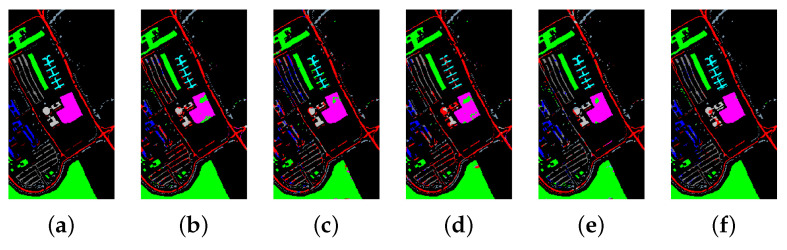
The classification maps of the University of Pavia. (**a**) Ground-truth map. (**b**)–(**f**) Classification results of 2-D-CNN, 3-D-CNN, Multi-3-D-CNN, HybridSN, and Transformer.

**Figure 15 sensors-21-01751-f015:**
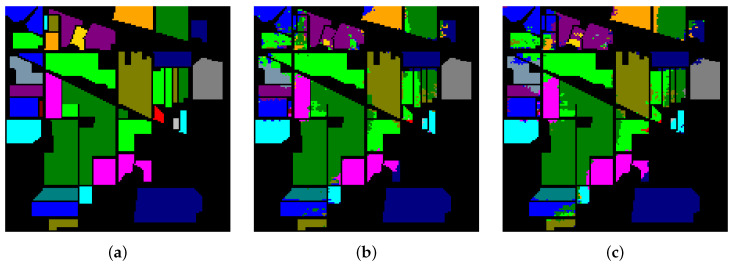
The classification maps of Indian Pines. (**a**) Ground-truth map. (**b**) Classification results of the Transformer without metric learning. (**c**) Classification results of the Transformer with metric learning.

**Figure 16 sensors-21-01751-f016:**
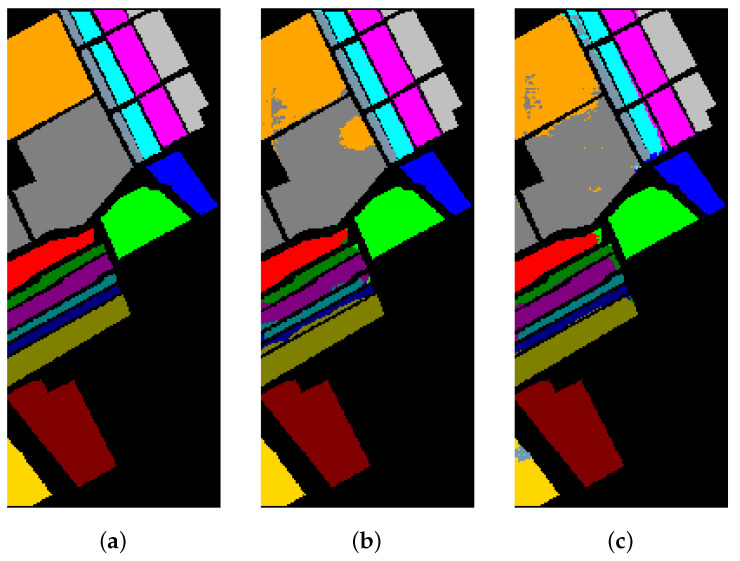
The classification maps of Salinas. (**a**) Ground-truth map. (**b**) Classification results of the Transformer without metric learning. (**c**) Classification results of the Transformer with metric learning.

**Figure 17 sensors-21-01751-f017:**
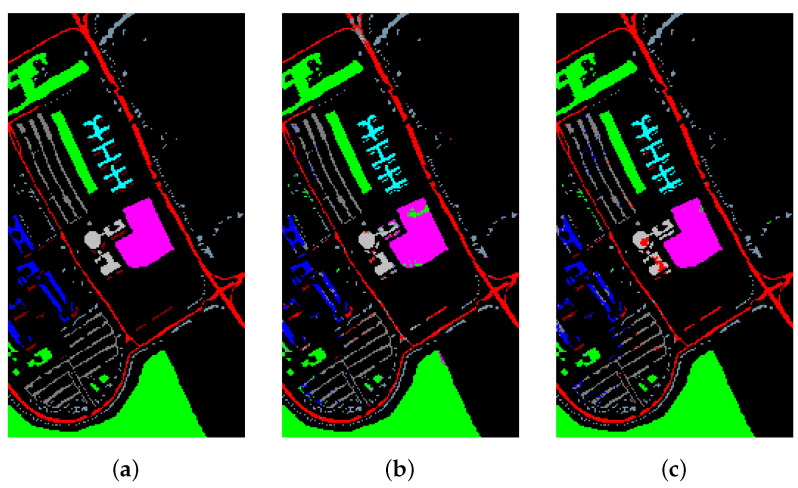
The classification maps of University of Pavia. (**a**) Ground-truth map. (**b**) Classification results of the Transformer without metric learning. (**c**) Classification results of the Transformer with metric learning.

**Figure 18 sensors-21-01751-f018:**
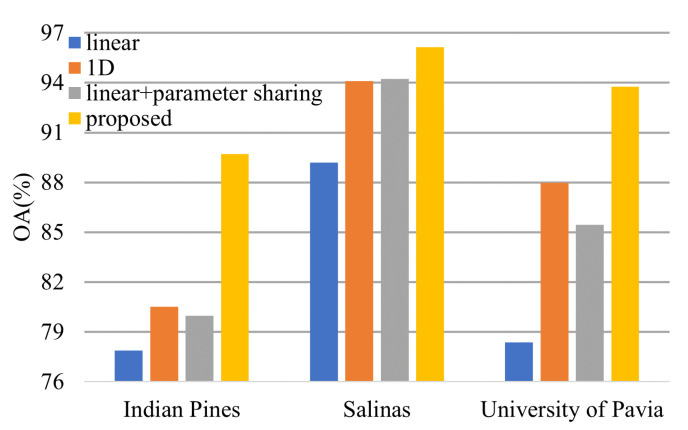
Effectiveness of the 1-D convolution and parameter sharing.

**Figure 19 sensors-21-01751-f019:**
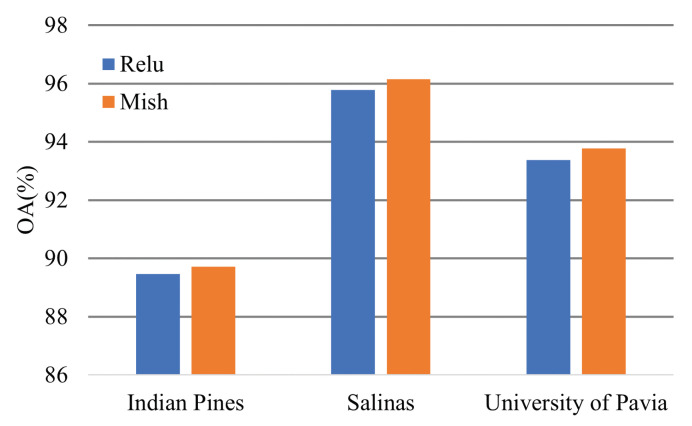
Effectiveness of the activation function.

**Table 1 sensors-21-01751-t001:** Training, validation, and testing sample numbers in Indian Pines.

Number	Name	Training	Validation	Testing	Total
1	Alfalfa	1	1	44	46
2	Corn-notill	42	42	1344	1428
3	Corn-mintill	24	24	782	830
4	Corn	7	7	223	237
5	Grass-pasture	14	14	455	483
6	Grass-trees	21	21	688	730
7	Grass-pasture-mowed	1	1	26	28
8	Hay-windrowed	14	14	450	478
9	Oats	1	1	18	20
10	Soybean-notill	29	29	914	972
11	Soybean-mintill	73	72	2310	2455
12	Soybean-clean	17	17	559	593
13	Wheat	6	6	193	205
14	Woods	37	37	1191	1265
15	Buildings-Grass-Trees-Drives	11	11	364	386
16	Stone-Steel-Towers	2	3	88	93
	Total	300	300	9649	10,249

**Table 2 sensors-21-01751-t002:** Training, validation, and testing sample numbers in Salinas.

Number	Name	Training	Validation	Testing	Total
1	Brocoli-green-weeds-1	8	8	1993	2009
2	Brocoli-green-weeds-2	14	14	3698	3726
3	Fallow	7	8	1961	1976
4	Fallow-rough-plow	5	5	1384	1394
5	Fallow-smooth	10	10	2658	2678
6	Stubble	15	15	3929	3959
7	Celery	14	14	3551	3579
8	Grapes-untrained	45	44	11,182	11,271
9	Soil-vinyard-develop	24	24	6155	6203
10	Corn-senesced-green-weeds	13	13	3252	3278
11	Lettuce-romaine-4wk	4	4	1060	1068
12	Lettuce-romaine-5wk	7	7	1913	1927
13	Lettuce-romaine-6wk	3	4	909	916
14	Lettuce-romaine-7wk	4	4	1062	1070
15	Vinyard-untrained	29	28	7211	7268
16	Vinyard-vertical-trellis	7	7	1793	1807
	Total	209	209	53,711	54,129

**Table 3 sensors-21-01751-t003:** Training, validation, and testing sample numbers in the University of Pavia.

Number	Name	Training	Validation	Testing	Total
1	Asphalt	33	33	6565	6631
2	Meadows	93	91	18,465	18,649
3	Gravel	10	10	2079	2099
4	Trees	15	15	3034	3064
5	Painted metal sheets	6	7	1332	1345
6	Bare Soil	25	25	4979	5029
7	Bitumen	6	6	1318	1330
8	Self-Blocking Bricks	18	18	3646	3682
9	Shadows	4	5	938	947
	Total	210	210	42,356	42,776

**Table 4 sensors-21-01751-t004:** Configuration of our proposed model variants.

Data set	Layers	Hidden Size	MLP size	Heads
Indian Pines	2	120	32	15
Salinas	2	75	32	15
University of Pavia	2	75	32	15

**Table 5 sensors-21-01751-t005:** Parameter summary of the proposed transformer model architecture over the Indian Pines data set.

Layer (Type)	Output Shape	Parameter
inputLayer	(30, 25, 25)	0
conv1d×25	(1, 120) × 25	632 × 25
2-layer transformer encoder	(1, 120)	132,064
layernorm	(120)	240
linear	(32)	3872
Mish	(32)	0
linear	(16)	528
Total Trainable Parameters: 152,504		

**Table 6 sensors-21-01751-t006:** Parameter summary of the proposed transformer model architecture over the Salinas data set.

Layer (Type)	Output Shape	Parameter
inputLayer	(15, 25, 25)	0
conv1d×25	(1, 75) × 25	302 × 25
2-layer transformer encoder	(1, 75)	55,564
layernorm	(75)	150
linear	(32)	2432
Mish	(32)	0
linear	(16)	528
Total Trainable Parameters: 66,224		

**Table 7 sensors-21-01751-t007:** Parameter summary of the proposed transformer model architecture over the University of Pavia data set.

Layer (Type)	Output Shape	Parameter
inputLayer	(15, 25, 25)	0
conv1d×25	(1, 75) × 25	302 × 25
2-layer transformer encoder	(1, 75)	55,564
layernorm	(75)	150
linear	(32)	2432
Mish	(32)	0
linear	(9)	297
Total Trainable Parameters: 65,993		

**Table 8 sensors-21-01751-t008:** Classification results of different models in Indian Pines.

No.	Training Samples	2-D-CNN	3-D-CNN	multi-3-D-CNN	HybridSN	Transformer
1	1	95.21	91.16	100.00	93.20	90.92
2	42	66.10	69.60	61.43	83.53	86.70
3	24	86.24	83.31	81.52	85.33	85.09
4	7	90.38	93.39	99.40	83.77	88.03
5	14	96.09	91.34	96.82	87.87	94.55
6	21	94.44	95.77	97.46	93.12	95.67
7	1	100.00	100.00	99.41	86.43	91.76
8	14	99.29	98.42	99.43	92.48	96.42
9	1	98.75	95.66	99.23	85.84	88.33
10	29	93.18	86.59	84.59	85.34	91.12
11	73	83.94	84.19	74.61	89.53	88.85
12	17	83.52	77.94	79.98	79.45	81.18
13	6	98.55	98.35	99.81	92.59	96.23
14	37	94.89	92.65	88.89	94.18	94.55
15	11	87.33	88.18	86.17	85.99	86.54
16	2	98.91	95.75	90.00	84.16	79.63
KAPPA		0.828±0.013	0.822±0.017	0.761±0.021	0.859±0.016	0.882±0.010
OA(%)		85.13±1.17	84.59±1.52	79.43±1.77	87.69±1.48	89.71±0.88
AA(%)		91.68±1.01	90.14±1.54	89.92±2.04	87.68±1.92	89.72±3.01

**Table 9 sensors-21-01751-t009:** Classification results of different models in Salinas.

No.	Training Samples	2-D-CNN	3-D-CNN	multi-3-D-CNN	HybridSN	Transformer
1	8	97.73	99.85	98.17	96.95	98.47
2	14	99.55	98.99	98.90	97.24	98.54
3	7	95.64	94.40	93.01	98.82	98.02
4	5	95.82	95.96	90.97	96.57	95.59
5	10	95.36	96.48	95.46	96.25	96.04
6	15	99.69	99.06	98.32	97.24	97.98
7	14	99.43	98.09	99.12	99.21	99.03
8	45	88.46	87.04	90.37	95.05	94.27
9	24	99.71	99.29	99.00	98.78	98.97
10	13	98.93	96.13	95.44	95.69	95.90
11	4	98.52	88.73	94.46	97.62	98.30
12	7	93.75	92.55	93.44	97.96	94.58
13	3	91.00	86.05	87.08	90.45	89.99
14	4	93.01	93.60	90.15	94.14	98.20
15	29	85.40	86.27	84.28	87.09	92.75
16	7	99.49	96.38	94.34	96.27	99.89
KAPPA		0.934±0.013	0.924±0.017	0.925±0.022	0.947±0.013	0.957±0.009
OA(%)		94.08±1.25	93.20±1.54	93.30±2.03	95.26±1.22	96.15±0.86
AA(%)		95.72±1.23	94.30±1.65	93.91±1.32	95.96±0.69	96.66±0.79

**Table 10 sensors-21-01751-t010:** Classification results of different models in the University of Pavia.

No.	Training Samples	2-D-CNN	3-D-CNN	multi-3-D-CNN	HybridSN	Transformer
1	33	72.50	70.35	71.81	83.17	89.98
2	93	94.77	95.83	96.62	96.64	96.89
3	10	85.90	62.36	73.75	70.79	88.56
4	15	95.62	77.50	84.48	84.67	94.82
5	6	97.54	98.49	96.05	94.76	92.43
6	25	97.06	96.47	94.88	94.94	98.06
7	6	97.78	80.24	83.94	80.61	88.01
8	18	77.08	64.31	69.62	71.55	84.98
9	4	87.23	69.38	71.26	85.06	93.89
KAPPA		0.848±0.010	0.795±0.029	0.825±0.028	0.851±0.049	0.916±0.014
OA(%)		88.75±0.77	84.72±2.21	86.92±2.10	88.90±3.63	93.77±1.06
AA(%)		89.50±1.75	79.44±3.61	82.49±3.44	84.69±3.09	91.96±1.86

**Table 11 sensors-21-01751-t011:** Parameter size of the five methods on the three hyperspectral data sets.

Network	Indian Pines	Salinas	PaviaU
2-D-CNN	176,736	165,936	98,169
	0.67 MB	0.63 MB	0.72 MB
3-D-CNN	1,018,476	771,516	447,374
	3.89 MB	2.94 MB	1.71 MB
multi-3-D-CNN	634,592	138,976	84,761
	2.42 MB	0.53 MB	0.32 MB
HybridSN	5,122,176	4,845,696	4,844,793
	19.54 MB	18.48 MB	18.48 MB
Transformer	152,504	66,224	65,993
	0.58 MB	0.25 MB	0.25 MB

**Table 12 sensors-21-01751-t012:** Flops of the five methods on the three hyperspectral data sets.

Network	Indian Pines	Salinas	PaviaU
2-D-CNN	11,708,240	5,995,040	5,927,280
3-D-CNN	162,511,650	83,938,540	83,614,405
multi-3-D-CNN	52,409,984	20,611,712	20,557,504
HybridSN	248,152,512	50,948,592	50,947,696
Transformer	5,294,912	1,988,762	1,988,538

**Table 13 sensors-21-01751-t013:** Running time of the five methods on the Indian Pines data set.

Data set	Algorithm	Training Time (s)	Testing Time (s)
**Indian Pines**	2-D-CNN	11.0	0.5
	3-D-CNN	54.1	4.26
	multi-3-D-CNN	56.23	5.10
	HybridSN	43.9	3.45
	Transformer	32.24	1.31

**Table 14 sensors-21-01751-t014:** Running time of the five methods on the Salinas data set.

Data set	Algorithm	Training Time (s)	Testing Time (s)
**Salinas**	2-D-CNN	6.0	1.9
	3-D-CNN	26.1	16.1
	multi-3-D-CNN	26.2	18.2
	HybridSN	13.9	7.5
	Transformer	13.8	4.6

**Table 15 sensors-21-01751-t015:** Running time of the five methods on the University of Pavia data set.

Data set	Algorithm	Training Time (s)	Testing Time (s)
**University of Pavia**	2-D-CNN	5.8	1.5
	3-D-CNN	26.2	12.7
	multi-3-D-CNN	26.2	14.2
	HybridSN	14.06	5.78
	Transformer	13.09	3.33

**Table 16 sensors-21-01751-t016:** Classification results of the transformer without metric learning and the transformer with metric learning.

	Indian Pines		Salinas	
No.	without	with	without	with
	Metric Learning	Metric Learning	Metric Learning	Metric Learning
1	99.16	90.92	98.47	98.47
2	85.41	86.70	98.69	98.54
3	84.68	85.09	96.58	98.02
4	85.99	88.03	93.18	95.59
5	92.39	94.55	96.34	96.04
6	94.99	95.67	98.25	97.98
7	72.42	91.76	99.10	99.03
8	95.71	96.42	93.83	94.27
9	79.72	88.33	99.39	98.97
10	89.07	91.12	96.13	95.90
11	89.17	88.85	98.46	98.30
12	79.96	81.18	94.71	94.58
13	96.65	96.23	91.32	89.99
14	95.47	94.55	97.16	98.21
15	89.88	86.54	92.36	92.75
16	89.52	79.63	99.93	99.89
KAPPA	0.876±0.016	0.882±0.010	0.955±0.009	0.957±0.009
OA(%)	89.26±1.39	89.71±0.88	96.02±0.88	96.15±0.86
AA(%)	88.76±3.33	89.72±3.01	96.49±0.96	96.66±0.79
	**University of Pavia**			
**No.**	**without Metric Learning**		**with Metric Learning**	
1	87.92		89.98	
2	96.80		96.89	
3	88.09		88.56	
4	92.32		94.82	
5	89.57		92.43	
6	96.60		98.06	
7	91.47		88.01	
8	85.53		84.98	
9	83.79		93.89	
KAPPA	0.905±0.019		0.916±0.014	
OA(%)	92.92±1.47		93.77±1.06	
AA(%)	90.23±2.16		91.96±1.86	

## Data Availability

The datasets involved in this paper are all public datasets.

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
