# Peer review of "A Lightweight 1-D Convolution Augmented Transformer with Metric Learning for Hyperspectral Image Classification"

_sensors, 2021, doi:10.3390/s21051751_

Round 1

Reviewer 1 Report

In the manuscript, the authors have presented a transformer driven model for HSI image classification. The approach is indeed now to HSI classification though already used for natural image classification and NLP. 

Overall, the authors have highlighted their novelty and provided evidence of the efficacy of the methods. However, there are several points that need to be clarified.

1) The authors should include a numerical evaluation of the computational complexity of methods and compare than instead of providing run-times only - which can often be misleading. 

2) The authors claim that the method is useful as it reduced convolutions, but again they have added a convolution layer which is counter-intuitive. 

3) The run time and accuracy of 2D CNN seems to be better than the proposed method. What is then the main rationale for proposing/using the new method. Please clarify with suitable justification. 

4) The authors claim that the method performs better for reduced datasets. Provide detailed results to support the claim with graphs showing how the network performance scale with changing dataset size (and compare it other state-of-the-art methods). A comparison graph with detailed explanation will be very helpful. 

Reviewer 2 Report

In this article, the authors, inspired by the Vision Transformer [18,19], propose a light transformer-based network for the classification of hyperspectral images. However, as the authors indicate, the classification performance of the transformer network on medium-sized datasets such as ImageNet was lower than that of ResNets of comparable size. So why they didn't compare their network with ResNets on the same HSI (large size) database?

While acknowledging that they have made numerous comparisons with leading methods based on convolutional neural networks, including 2-D CNN [23], 3-D CNN [24], multi-scale 3-D CNN [25] and hybrid CNN,

The presentation of the paper can be largely improved. The paper is too long. There are
several sections, which presents known things. The author spends too much content to
introduce the common knowledge, e.g. section 2. I suggest the author to reduce section 2,
and put more emphasis on the illustration and analysis of your contribution.
I. The authors make certain claims during the discussion of the experimental results
that either needs more clarification/explanation or needs some referencing.
II. II. Some grammatical errors in the paper. Some choice of words are inappropriate
to convey the meaning of the sentence, the reviewer suggest some revision be
made to clarify the sense and meaning of the sentences. Some sentences are too
complex (running at 30-40 words in one sentence) that makes it hard to grasp
their meaning, making the sentences vague. The authors should use comma in
sentences sparingly. Concise but nevertheless complete sentences are
encouraged, rather than comma separated phrases formed into compound
sentences. The authors should also make an effort to explain the entities in the
Figures mentioned for added clarity and readability of the paper.

Reviewer 3 Report

The paper brings an interesting contribution about the use of transformers to hyperspectral images classification. It also brings a good set of experiments. However, the discussion and ablation about the results obtained should be improved considering the comments which follows. The description of the proposed methodology, whose presentation I considered that is lacking, for instance, important mathematical equations about transformers should be improved as well.

On page 1, in line 25, please remove in tasks from the sentence “in tasks in computer vision tasks,”

There is some space characters lacking before open parentheses in lines 30 and 32.

On page 1, in line 32, perhaps it is lacking the adjective “restricted” of RBM in full name.

On page 1, in line 34, I consider one-dimensional a little accurate therm for the context of hyperspectal image, since even from one single pixel we have several features.

On page 2, in line 48, it would be better writing NLP in full.

On pages 2, 3 and 4, the parameters summary of models architectures over the three datasets fits better in the beginning of the experiments (section 3). At this point, it would be better making a generic, parametric presentation of the architecture instead.

Since transformer is relatively recent architecture, section 2 should be extended presenting the model in more detail. It lack also of citations of the papers that inspired the proposed architecture.

In section 3.2, please describe the meaning of the numbers after the plus-minus signs which appear on tables 8, 9, 10 and 15.

On page 9, in line 156 please include a bibliographic reference for hybridSN.

In general therms, I recommend that the results discussion be extended, considering more aspects about the compared networks, about the spatial (not only numeric) accuracy of generated thematic maps as well as classes spectral similarity.

I would also compare the obtained numeric (accuracy and Kapppa) results to the ones provided by the best state of the art approaches presented in the literature.

Round 2

Reviewer 3 Report

Dear authors to the best of my knowledge and belief I recommend this paper be accepted in the present form.

Regards,

Author Response

Thank you very much for your helpful suggestions and valuable comments.